# Improving the Safety and Security of Software Systems by Mediating SAP Verification

**Maram Fahaad Almufareh *** and **Mamoona Humayun ***

Department of Information systems, College of Computer and Information Sciences, Jouf University, Sakaka 72347, Saudi Arabia
* Correspondence: mfalmufareh@ju.edu.sa (M.F.A.); mahumayun@ju.edu.sa (M.H.)

**Abstract:** Security and performance (SAP) are two critical NFRs that affect the successful completion of software projects. Organizations need to follow the practices that are vital to SAP verification. These practices must be incorporated into the software development process to identify SAP-related defects and avoid failures after deployment. This can only be achieved if organizations are fully aware of SAP verification activities and appropriately include them in the software development process. However, there is a lack of awareness of the factors that influence SAP verification, which makes it difficult for businesses to improve their verification efforts and ensure that the released software meets these requirements. To fill this gap, this research study aimed to identify the mediating factors (MFs) influencing SAP verification and the actions to promote them. Ten MFs and their corresponding actions were identified after thoroughly reviewing the existing literature. The mapping of MFs and their corresponding actions were initially evaluated with the help of a pilot study. Mathematical modeling was utilized to model these MFs and examine each MF's unique effect on software SAP verification. In addition, two case studies with a small- and a medium-sized organization were used to better understand the function these MFs play in the process of SAP verification. The research findings suggested that MFs assist software development organizations in their efforts to integrate SAP verification procedures into their standard software systems. Further investigation is required to support the understanding of these MFs when building modern software systems.

**Keywords:** mediating factors; security; performance; verification; non-functional requirements

## 1. Introduction

The relevance of software systems to modern civilization generates prodigious concerns about several essential quality attributes. Software developers commonly describe such qualities as NFRs. Security, performance, reliability, maintainability, usability, and scalability are all defined by NFRs [1,2]. They constrain the system's design across backlogs. NFRs, or system qualities, are as vital as functional requirements. They ensure that the system is useful and effective. Systems that fail to fulfill any of these requirements may fail to meet internal business, user, or market expectations [3,4]. These NFRs may result in substantial legal penalties if they are not adhered to. All NFRs are equally significant and must be addressed; however, a single study will not be able to handle all of them. As a result, the goal of this research was to discuss only two NFRs, namely security and performance. We must understand the MFs for SAP verification to effectively address SAP needs.

Security is defined as "the condition of being free from danger or peril" [5]. The use of methods/techniques to analyze, mitigate, and defend software systems against vulnerabilities is known as software security. These methods guarantee that software continues to operate and is protected against threats [5,6]. Security considerations must be made at every level of the software development process. The primary objective is to find faults and problems as early as feasible. Software performance is another critical

component of daily operations, yet many of the issues arise from the software's actual code, not from external variables. Software performance difficulties are often introduced by engineers who were unaware that they would exist. In other cases, problems are not obvious because they are not caused by the code itself but by how the code responds to something else [7]. Although security and performance are two different NFRs, they are somehow related to each other. Security is a performance indicator that is affected by threats that impact the performance of individual components of software during service rendering [8]. Both security and performance illustrate the software's efficiency, implying that performance and security are indicators of the software's development level.

Even though various technologies facilitate software development, this is a human-centered activity that is prone to errors. Software systems must adhere to SAP; to achieve this, software development companies include quality assurance activities throughout the product life cycle to analyze these qualities, hence avoiding SAP difficulties after software release [9]. The general motive for this research effort was as follows: (1) understanding the significance of SAP requirements for software systems and the need to include SAP verification in the development process, and (2) identifying the SAP MFs and their role in software success. Security and performance verification are actions that seek flaws in these two distinct quality axes. Different verification procedures and methodologies may be employed alone or in combination, each with its own advantages and drawbacks for SAP verification.

Various SAP verification methods are out there, but software systems still have flaws. Performance difficulties are a major concern in several businesses, such as telecoms, and attacks are also increasing rapidly on various systems, such as on news-reporting systems [10]. Some of the possible causes for this issue are (1) inefficient verification methods, (2) software businesses failing to apply adequate verification procedures, or (3) a perceived split between academia and industry. Automated attack scripts, the availability of attack knowledge, and worldwide interconnectedness have made it simpler to attack software systems. To better address the SAP requirements of the software, there is a need to understand the MFs of SAP and the ways to address these MFs efficiently.

## 1.1. Research Approach and Contribution

The study's overall goal was to discover the MFs for SAP verification and their influence on software success. The following were the specific objectives:

- A thorough literature review to extract the MFs for software SAP;
- Identifying the verification practices for SAP and the techniques used to implement these practices;
- Analyzing the impact of identified verification practices on SAP individually;
- Validate the findings with the help of case studies.

## 1.2. Paper Organization

The next part delves into software SAP verification criteria based on the current research in the area under investigation. Section 3 discusses some existing studies to provide the current state-of-the-art. Section 4 extracts the mediating factors for SAP verification from the existing literature. Section 5 elaborates on the proposed methodology. Section 6 presents the case study findings and is followed by Section 7, which presents the results and recommendations for future research. Table 1 shows the list of abbreviations used in this study for a better understanding.

**Table 1.** List of abbreviations used.

| Abbreviations | Full Form |
| --- | --- |
| IAST | Interactive application security testing |
| MF | Mediating factors |

**Table 1.** *Cont.*

| Abbreviations | Full Form |
| --- | --- |
| MS-SDL | Microsoft security development lifecycle |
| NFR | Non-functional requirements |
| QA | Quality assurance |
| RE | Requirement engineering |
| ROI | Return on investment |
| SAMM | Software assurance maturity model |
| SCA | Static code analysis |
| SDLC | Software development lifecycle |
| SSD | Secure software development |
| SSDLC | Secure software development lifecycle |
| SSE-CMM | System security engineering capability maturity model |
| SAP | Security and performance |
| SVTs | Security verification techniques |
| ZAP | Zed attack proxy |

## 2. Background

SAP are key NFRs that need to be paid attention to for achieving a quality software product. In this section, we will analyze both these NFRs' roles in software quality and the techniques used to ensure it.

### 2.1. Software Security Verification

Software security has always been an afterthought during the testing phase of development. In contrast, modern approaches, such as agile, include continuous testing throughout all stages of the SDLC. Every day, hackers and cybercriminals are coming up with new methods to take advantage of software flaws. Security should be a priority throughout the SDLC, allowing developers and stakeholders to identify and resolve possible security concerns early on in the process [11,12]. Therefore, it is necessary to adopt the concept of SSDLC, as shown in Figure 1.

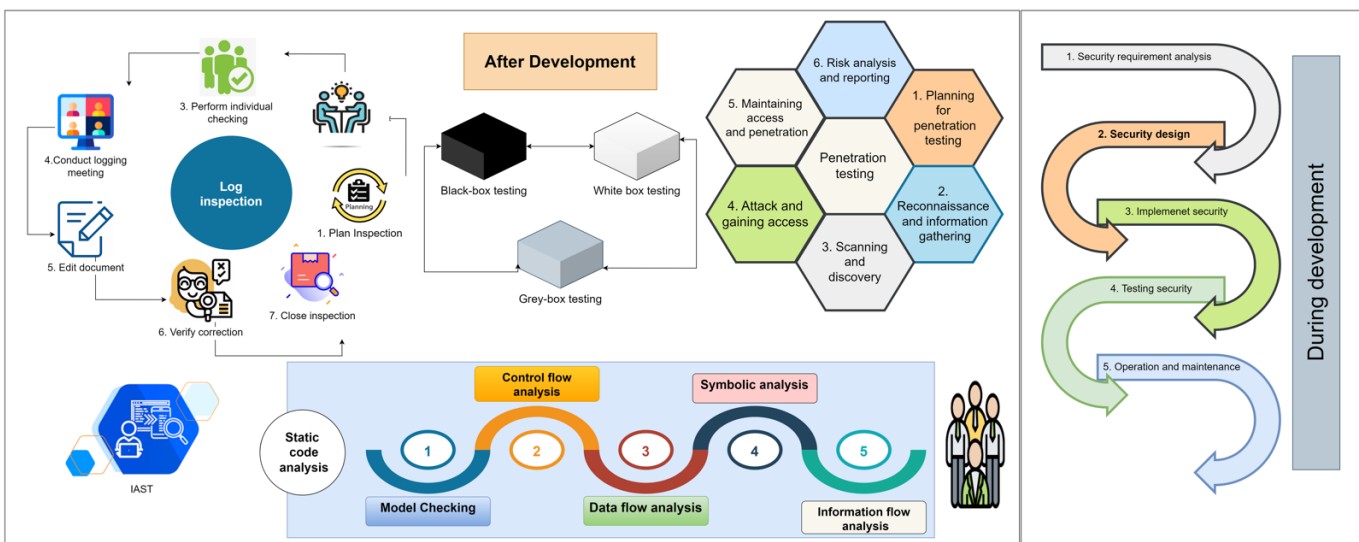

**Figure 1.** An insight into security verification.

The security verification of software, also known as security assurance, spins around the following main techniques, as shown in Figure 1:

- Log inspection;
- Static code analysis;
- Profound penetration testing before release;
- Interactive application security testing (IAST).

The complete concepts of security adoption and verification are better elaborated in Figure 1. According to Figure 1, organizations need to adopt SSDLC practices during development, which include security requirement analysis, security design, security implementation, security testing, and operation and maintenance. Once software development is complete, it needs security verification before release; this includes applying essential security verification techniques, as mentioned above. Below, we describe the concept of SSDLC and key SVTs.

### 2.1.1. SSDLC

SSDLC involves incorporating security best practices into an existing SDLC for achieving secure software. The SSDLC procedure demands concerted labor at each level of the SDLC, from requirement collection through deployment and maintenance [13]. SSDLC involves a mind change on the part of the development team, focusing on security at each step of the project rather than merely on functionality. Security vulnerabilities may be addressed in the SDLC pipeline well before deployment to production with a dedicated effort. This minimizes the possibility of detecting security vulnerabilities in software and seeks to lessen the consequences if detected [14]. The purpose of SSDLC is not to replace conventional security checks but rather to integrate security into software development duties and empower them to build safe apps from the start.

### 2.1.2. Security Verification Technique

Security verification refers to the technical verification of the application before deployment. Security verification aims to identify security breaches and to verify that the developed software meets security requirements [10,15]. Security verification is also necessary to maintain a business's reputation and prevent sensitive data loss. In order to establish security requirements, organizations first need to understand the threat and risk modeling technique/process. Figure 2 elaborates on the threat-modeling process for better identification of risks and their associated severity.

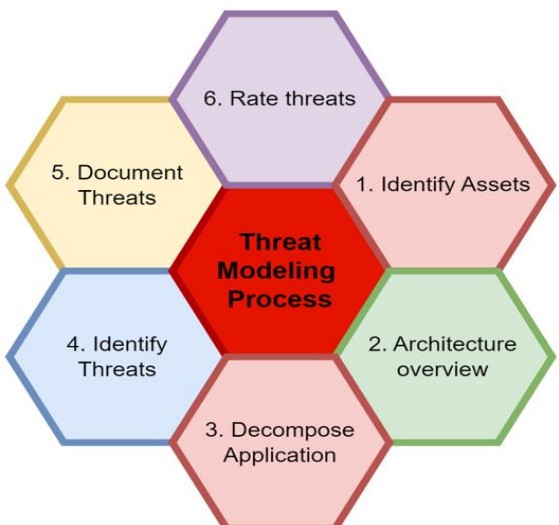

**Figure 2.** Threat-modeling process.

Below, we briefly discuss the commonly used security verification techniques.

Log Inspection

This SVT is used to discover software failures and faults. Log inspection is a systematic process that consists of seven activities, as shown in Figure 1. These activities include planning the inspection, holding kick-off meetings, performing individual checking, conducting log meetings, editing the document, verifying corrections, and inspection closure [8,10,16]. The verification team is responsible for performing log inspections.

Penetration Testing

The penetration test is often performed using a failure-based approach, with test cases designed to investigate known flaws found in popular security vulnerability repositories. Another penetration test approach is experience-based, in which a security expert plays the role of a malicious user attempting to access the system [17,18]. It is a systematic process consisting of six steps, as mentioned in Figure 1. Penetration testing is performed either manually or with the help of automated tools. Arachni, SQL Injection ME, OWASP ZAP, XSS ME, Meta Exploit, NMAP, Burp Suite, and Whatweb are the most-often-used tools for automated penetration testing.

Static Code Analysis

Using static code analysis (SCA) before executing a program is a debugging method. To achieve this task, one must compare a code set to one or more sets of coding rules. It resolves source code flaws that might lead to vulnerabilities. SCA may be performed manually by examining each line of code or automatically using automated techniques, but primarily, SCA is performed in an automated way [10,19]. HP Fortify SCA and Brakeman are common standard tools used for static code analysis.

Interactive Application Security Testing (IAST)

IAST solutions assist firms in uncovering and managing security risks associated with runtime application vulnerabilities. IAST uses software instrumentation to monitor an application's behavior and performance. IAST solutions put agents and sensors in operating applications and analyze all application interactions to find vulnerabilities in real-time.

*2.2. Software Performance Verification*

The effectiveness of a software system in terms of time restrictions and resource allocation is measured by its performance. Response time, throughput, and utilization are the traditional performance measures. The response time is the time it takes a job from start to finish to transit a certain route inside the system. Throughput is the number of tasks a certain section of the system can carry out in a given time. The proportion of time that a certain element of the system is actively functioning for is referred to as utilization [20]. The success criteria for the testing procedure must first be established before performance testing can begin. Target metrics become the main focus when preparing and creating performance test cases. Therefore, measurements constitute the basis for performance testing. Monitoring the appropriate indicators may assist in identifying areas that need greater attention and determining how to improve them [21]. Figure 3 provides the common performance testing types and a complete step-by-step performance verification process.

2.2.1. Performance Verification Techniques

To determine if performance is sufficient, an organization must first set milestones. Then, it should measure the metrics that fall within these milestones and estimate the outcome by comparing the actual and projected results. Metrics help an organization in the following ways:

- They serve as a foundation for the testing;
- They aid in the tracking of a project's progress;
- A QA team may use metrics to describe and quantify a problem to discover a solution;

- Metrics tracking over time helps the organization to compare test results and evaluate the effect of code changes.

Which performance metrics do a QA team need to track? It depends on the nature of the software under testing. Some commonly used parameters in performance metrics include response time, request per second, user transactions, error rate, wait time, average load time, peak response time, concurrent users, throughput, CPU utilization, memory utilization, and total user sessions. Performance metrics need to be used very carefully; some principles of using performance metrics effectively are listed as follows:

- Assemble an exhaustive set of performance criteria by first identifying the specific business goals of your customer;
- A feature must be given an appropriate measure of success for it to be considered successful;
- If a software user receives a high performance, reliability, and functionality level, metrics should reflect this;
- Track metrics over time by doing repeated performance tests;
- Run each software item through its paces one at a time. Perform database, service, and other checks.

Once the milestone is set, the organization may apply various manual and automated ways to evaluate software performance. Below, we discuss some of the standard techniques of performance verification.

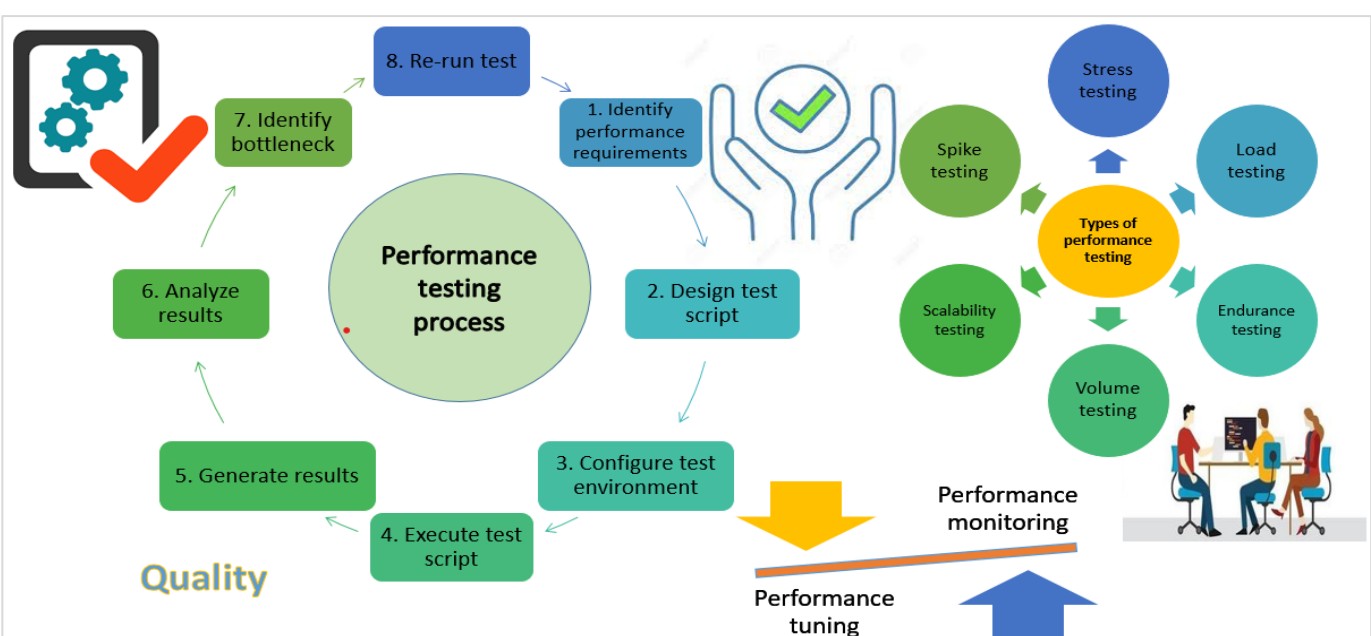

**Figure 3.** An insight into performance verification.

Log Inspection

Log inspection or monitoring helps to detect malicious behavior, collect events across different platforms, inspect error and informational events, and create and maintain audit trails [22]. Thus, it is a valuable measure of evaluating software performance.

Response Time Test

The response time verification technique measures the time between a user's request and a server's, application's, website's, or device's response. For example, if a user asks an application to load a particular web page, the response time is how long it takes the program to respond. Response time assists developers in assessing if software and websites are functional and responsive enough to be delivered as a final product [23]. A faster response

time ensures customer satisfaction and higher ratings for software or websites. Slower response times may indicate faults, hardware difficulties, or connection concerns. The response time verification consists of four main steps: identifying response time parameters, performing response time tests, recording the findings, and identifying errors or successes.

Resource Consumption Test

Another crucial performance verification technique of an application's performance, in addition to log inspection and response times, is resource consumption. A performance test is a technique to keep track of how active particular computer system resources are [22,24]. The critical consideration of resource consumption includes four main elements: CPU, memory, disk, and network.

Throughput

The number of transactions created over time during a test is referred to as "throughput". It is also known as the maximum capacity that an application can support. Before beginning a performance test run, it is typical to set a throughput target, requiring the application to handle a certain number of requests per hour [25]. The formula for the throughput is as given in Equation (1)

$$\text{Throughput} = \frac{(\text{number of requests})}{(\text{total time})} \tag{1}$$

After obtaining an overview of related terms, in the next section we provide the existing studies on software SAP verification in order to obtain a picture of the current state-of-the-art.

## 3. Related Work

In this section, we will explore the existing studies to extract the mediating factors (MFs) that impact the SAP of software so that we may analyze the impact of these MFs on software development.

Victor et al. in paper [8] identified eight MFs for SAP, which include organizational awareness, suitable requirements, cross-functional team, verification environment, support tools, verification methodology, verification planning, and reuse practices. Rapid reviews were used in this study to identify actions for promoting each identified MF. A survey with 37 software practitioners was conducted for the classification of MFs and their corresponding actions. The MFs, according to this research, may be regarded as crucial considerations in helping software development firms to implement/improve SAP verification operations in routine software.

The paper [10] summarizes the findings of a case study conducted on three Brazilian software companies. The report analyzes the current state of practice for SAP verification operations. Furthermore, the results are explained in the form of hypotheses, which constitute suggestions relevant to such actions. In general, there is a growing understanding of the relevance of software system SAP and verification operations. However, there is a lack of understanding about how verification should be carried out. This study also offers suggestions for improving SAP verification operations.

The paper [26] presents an evidence-based body of knowledge that characterizes the most important NFRs for software systems and appropriate testing methodologies for evaluating these requirements. The study further delves into the SAP verification procedures used in software development companies and the MFs that influence decision-making. In addition, MFs for SAP verification activities, as well as methods to promote them, are discussed. The conclusions of this study are evidence-based since they are based on several research methodologies and in-person observations in the software business.

According to the paper [27], several methods have been developed throughout the years to make the process of developing and deploying secure software easier, with varying degrees of success. To obtain a better understanding of the underlying difficulties, this

article explains and assesses a variety of SCA approaches and tools using an example that highlights some of the most common software security concerns. The latter issue may be handled by taking into consideration a strategy that allows for the identification of security characteristics and their translation into security rules that can be tested against the criteria of security standards. This would assist developers throughout the software development lifecycle and would help to ensure that a product is compliant with security standards.

The study [28] proposes an architectural framework for analyzing the performance deterioration of software caused by security measures. It introduces a collection of UML models that describe security mechanisms and may be used with performance-annotated UML application models to build SAP-critical systems. Model composition enables alternative security solutions to be introduced on the same software architecture, assisting software architects in finding acceptable security solutions while still fulfilling performance requirements. The experimental findings justify the suggested technique by comparing a model-based assessment of a software architecture for managing cultural assets with values seen in the actual implementation of the system.

### 3.1. Existing SSDLC Models

In order to better understand SSDLC, we need to analyze the existing SSDLC models. Below, we provide the details of common SSDLC models and their comparison.

#### 3.1.1. SSE-CMM

Based on software engineering CMM, the SSE-CMM is a process-oriented paradigm used to create safe systems. The SSE-CMM is divided into processes and stages of maturity. In general, the procedures specify what the security engineering process must do, and the maturity levels classify how well the process achieves its objectives. To use the SSE-CMM, current processes are simply analyzed to see whether basic processes have been satisfied and what maturity levels they have attained. The same procedure may assist a company in identifying security engineering procedures that they may need but do not already have in place.

#### 3.1.2. Microsoft SDL

Microsoft's SDL incorporates stringent security standards, industry-specific tools, and required procedures into the creation and maintenance of all software products. By following the SDL methods and criteria, development teams may utilize MS-SDL to produce better secure software with fewer and less-serious vulnerabilities at a lower development cost. There are seven phases in MS-SDL comprising the five SDLC core stages and two auxiliary security tasks. To make sure all security and privacy needs are correctly met, each of these steps includes necessary checks and approvals. To make sure the core phases are implemented correctly and that software is kept safe after deployment, the two supporting security activities, training and response, are carried out before and after the core phases, respectively.

#### 3.1.3. SAMM

SAMM is an open framework designed to assist businesses in developing and implementing a software security strategy appropriate to the unique threats they face. SAMM's tools may be used to assess an organization's current software security procedures, create a well-rounded software security program in measurable steps, verify the efficacy of a security assurance program, and quantify security-related endeavors. Because of its adaptability, SAMM may be used by companies of all sizes and in any development methodology. This model's flexibility means it may be used throughout an enterprise, inside a business unit, or even on a standalone project.

After discussing the three commonly used existing SSDLC models, below, we provide a comparison of these models in Table 2.

**Table 2.** Comparison of existing SSDLC.

| Security Practices | SSE-CMM | MS-SDL | SAMM |
|---|---|---|---|
| Physical security | Yes | No | No |
| Logical security | Yes | No | No |
| Definition of security requirements | Yes | Yes | Yes |
| Secure configuration management | Yes | No | No |
| Following all applicable laws, policies, and procedures | Yes | No | Yes |
| Threat modeling | Yes | Yes | Yes |
| Risk analysis | Yes | Yes | Yes |
| Security architecture | Yes | Yes | Yes |
| Security training and awareness | Yes | Yes | Yes |
| Secure design | Yes | Yes | Yes |
| Source code analysis | No | Yes | Yes |
| Vulnerability analysis | Yes | Yes | Yes |
| Security verification | Yes | Yes | Yes |
| Vulnerability management | Yes | Yes | Yes |
| Secure development techniques and applications | Yes | Yes | Yes |
| Security in an active operating environment | Yes | Yes | Yes |
| Secure integration with peripheral | Yes | Yes | Yes |
| Secure delivery | Yes | No | Yes |

The analysis of the literature review and comparison of existing security models showed that security is one of the important concerns discussed by many researchers and security organizations. However, we did not find studies that provide solutions for both security and performance or uncover the MFs that affect SAP.

Security and performance are two essential NFRs that play critical roles in software quality and effective deployment. The existing research either discusses security or performance; further, they provide general solutions for SAP improvement. Security and performance are complementary requirements; therefore, there is a need to identify the factors that mediate the process of SAP verification. This demonstrates the importance of identifying MFs for software SAP as well as evaluating the functions of these MFs in SAP verification. To fill this gap, this research first identified the MFs for SAP from the existing scientific and grey literature. Then, we evaluated the impact of these MFs on software SAP.

## 4. Mediating Factors for SAP Verification

The SAP-moderating factors may be seen as suggestions that must be taken into account during SAP verification. To identify these MFs, we searched throughout the formal and grey literature and identified the following ten factors that serve as MFs for SAP verification.

### 4.1. SAP Verification Planning and Methodology

SAP verification is usually not well planned, due to which the software team must reprioritize verification activities multiple times, increasing time and effort. The planning phase includes requirement prioritization, dependency identification, and time and budget for performing SAP verification activities. Further, a well-planned and transparent methodology for performing SAP verification should be defined to smoothly execute the verification process [8,10,29].

### 4.2. Suitable Environment for SAP Verification

Organizations need to configure the environment to make it suitable for performing SAP verification. A proper environment is inevitable for verification since it comprises both the settings of the infrastructure responsible for system operation as well as the configuration of the system itself [8,10,29].

### 4.3. Organizational Support

Organizational views of SAP verification often harm this activity. Among these perceptions is the notion that SAP verification wastes resources. Typically, businesses worry about their systems' security and effectiveness only when something goes wrong. Another time when companies prioritize SAP is prior to a significant release when a security or performance failure might negatively affect the organization's image [26,29]. This shows that organizations should prioritize SAP verification to produce high-quality software and to maintain their reputation.

### 4.4. Defining Complete SAP Requirements/Document Security Policy

A lack of SAP requirements precludes the verification from fulfilling its intended function (that is, determining if the program fulfills its requirements) since it is impossible to know whether the verification findings are valid without an oracle. Incomplete requirements also strain other teams (e.g., analysts, architects, and developers) since the verification team must constantly contact them [8,29,30]. Therefore, SAP requirements should be completely specified during the requirement-engineering phase along with test cases.

### 4.5. Software Team Awareness

SAP verification is not only the job of the QA team; instead, it is the responsibility of the whole development team. Programmers must be aware of the inherent flaws in the technology employed in software development and the coding patterns that lead to errors. They must be provided training for this so that they may avoid using faulty technology and increase their ability to write error-free code. Managers must also receive training to see SAP verification efforts as an investment rather than an expenditure. Furthermore, training may expose managers to the issues that a software product with inadequate SAP might cause, e.g., loss of customers and high infrastructure costs [8,19,30,31]. Consequently, managers will see the value of adding SAP verification operations in the product life cycle and allocating money and time for them.

### 4.6. Cross-Functional Teams

A single team does not carry out verification operations in isolation. It necessitates collaboration between many teams with a variety of abilities. Due to the requirement of installing servers, restoring database backups, and dealing with certain low-level technologies, infrastructure, database, and technical teams must collaborate. The identification of the technologies utilized to construct the program and how such technologies may impact the verification findings requires cross-team collaboration [10,29,32]. Meetings involving the performance architect, domain experts, marketing stakeholders, and developers may regularly assist in enhancing team relationships and, as a result, SAP verification.

### 4.7. Suitable Techniques of SAP Verification/Tool Support

Ad-hoc verification obstructs the creation of criteria for selecting test cases and the establishment of a definition of done. A suitable verification approach naturally provides these requirements. For example, using tools without understanding exactly what approach is being used creates uncertainty regarding the tool's detection capabilities. Additionally, the quantity of false-positive flaws found by security tools might be an issue; analyzing all the problems reported involves significant additional labor [12,16,29]. Therefore, selecting suitable techniques for SAP verification is necessary for efficiently performing SAP.

*4.8. SAP Monitoring and Audit/Define Measure*

Monitoring verification operations helps to verify that the software development team members adhere to SAP's best practices. Additionally, it enables the team to identify unusual activity, such as privilege misuse and the impersonation of another user. The software development team must identify essential metrics that are meaningful and relevant to the project's requirements [22,26,29,33]. Metrics that are well-defined aid in assessing the SAP position over time.

*4.9. Allocation of Time and Budget to SAP Verification*

SAP verification operations are often neglected when a development project's time-frame or budget is shortened. This may affect the system's quality since the product owner may terminate the verification process before all activities are completed, reducing the verification coverage [29,34,35]. As a result, the duration and budget for SAP operations should be determined at the start of the project when the SAP verification policy is documented.

*4.10. Encourage Reusability*

Reusing knowledge and artifacts enables SAP verification processes to be more agile. Reusing functional test cases in performance tests may be advantageous since they reflect real-world use situations. Additionally, test cases for earlier systems' parameters may be adapted, decreasing effort and time. For example, the necessary reaction time for a scenario may be established using a comparable production system scenario [10,29,36].

Now, we analyze the impact of the above-mentioned MFs/recommendations/best practices on SAP verification activities in the subsequent sections.

**5. Proposed Methodology**

This study aimed to optimize software SAP verification by identifying and analyzing the factors that serve as MFs and indirectly help to improve SAP verification. Before proceeding toward finding the impact of the identified MFs on SAP verification, there is a need to identify the actions that may promote the identified MFs. Table 3 lists the MFs of software SAP and actions/measures for promoting these MFs from the existing literature on software SAP.

**Table 3.** MFs and the corresponding actions to promote these MFs.

| MFs | MFs Detail | Actions | Actions/Practices to Promote MFs |
|---|---|---|---|
| MF1 | SAP verification planning and methodology | A1 | Use of tools to guide planning |
| | | A2 | Define acceptance criteria |
| | | A3 | Define systematic verification methodology |
| | | A4 | Choose a methodology that meets stakeholder demands and avoids sloppy methods |
| | | A5 | Create and update procedures that fit the defined methodology and organizational needs |
| MF2 | Suitable environment for SAP verification | A1 | Execution environment simulation with virtualization technology |
| | | A2 | Keeping the verification team up to date on the latest technology utilized in the project |
| | | A3 | Assembling test agents in a virtual environment |
| | | A4 | Using automated verification |
| | | A5 | Simulating a software's actual behavior by mimicking actions |

**Table 3.** *Cont.*

| MFs | MFs Detail | Actions | Actions/Practices to Promote MFs |
|---|---|---|---|
| MF3 | Organizational support | A1 | Keeping the software team updated on SAP |
| | | A2 | Educating the customer on the current situation of SAP |
| | | A3 | Regular security meetings |
| | | A4 | Resource assignment for SAP verification |
| | | A5 | External audit to support SAP verification |
| | | A6 | Promoting training |
| MF4 | Defining complete SAP requirements/document security policy | A1 | Using techniques for the identification of SAP requirements |
| | | A2 | Involving the verification team in the RE process |
| | | A3 | Motivating the verification team to examine the testability of the specifications |
| | | A4 | Involving the RE team in SAP verification activities |
| MF5 | Software team awareness | A1 | Promote training |
| | | A2 | Keeping the development team well-informed about SAP |
| | | A3 | Regular meetings |
| MF6 | Cross-functional teams | A1 | Building a team with multiple skills |
| | | A2 | Invest in training to improve team qualification |
| | | A3 | Promoting interaction and communication |
| | | A4 | Encourage agile/scrum |
| | | A5 | Hire qualified people |
| | | A6 | Leader swapping |
| MF7 | Suitable techniques of SAP verification/tool support | A1 | Identification of suitable tools |
| | | A2 | Encourage the use of freely available tools |
| | | A3 | Allow the verification team to suggest tools |
| | | A4 | Training of new tools |
| | | A5 | Support from the tool vendor |
| | | A6 | Using best practices toolset |
| | | A7 | Develop a culture of tool usage |
| MF8 | SAP monitoring and audit/define measure | A1 | Define key metrics relevant to project needs |
| | | A2 | External audit |
| | | A3 | Monitoring plans |
| [MF9 | Allocation of time and budget to SAP verification | A1 | Budget forecasting |
| | | A2 | Categorizing budget |
| | | A3 | Budgetary control |
| | | A4 | Keep customers informed about the benefit of SAP verification |

**Table 3.** *Cont.*

| MFs | MFs Detail | Actions | Actions/Practices to Promote MFs |
|---|---|---|---|
| MF10 | Encourage reusability | A1 | Keep track of prevalent faults and utilize test cases to discover their failures. |
| | | A2 | Using other comparable systems' expertise to define the needs |
| | | A3 | Reusing functional test cases that reflect real-world use |
| | | A4 | Adapting settings from related systems test scenarios |
| | | A5 | Vulnerability mapping by domain to identify situation-specific vulnerabilities |
| | | A6 | Design real-time scenarios |

Once the MFs and the corresponding actions to promote these MFs were defined, a methodology was proposed to evaluate the SAP verification process. Algorithm 1 describes the overall working of the proposed methodology.

According to Algorithm 1, the organization first needs to choose the SSDLC. The choice of SSDLC depends on various factors, including the nature of the project, budget, and time constraints. Once the methodology is defined and the requirements for the SSDLC are established, the MFs for SAP need to be identified. The proposed methodology identified ten MFs from both the scientific and grey literature. Next, what actions are required to accomplish these MFs need to be determined. In our case, we extracted the MFs and their corresponding actions from the existing published literature to ensure their correctness and effectiveness [8,26,29,35]. Once the MFs and their corresponding actions are identified, the next step is to prioritize these actions according to the nature and complexity of the project. For prioritization, every action is assigned a numeric weightage based on its importance for the project. Next, the organization needs to evaluate each action's impact on MFs, as shown in Equation (5). If the impact of an action on the identified MF is high, it will be adapted; otherwise, it may be ignored. Below, we present the proposed methodology using mathematical modeling. The idea of mathematical modeling was taken from [37,38].

The objective function of this study was to optimize *SPV*, as mentioned in Equation (2):

$$OF = Max(SAP) \therefore Improving\ SPV \tag{2}$$

$$\text{where } SPV = f(MF_1,\ MF_2, MF_3 \ldots \ldots MF_{10}) \tag{3}$$

$$MF_i = \sum_{j=1}^{n} A_j W_j \tag{4}$$

where $w_j$ is the weight assigned to each $A_j$ and $A_j$ represents the actions mentioned in Table 3

To maximize the overall SAP, appropriate actions need to be taken for improving each MF; this can be accomplished by satisfying Equation (5):

$$MF_i = \frac{1}{n} \sum_{j=1}^{n} A_j W_j \geq Tv\ (MF_i) \tag{5}$$

where $Tv$ is the threshold value corresponding to each $MF_i$ depending on the nature of the project and other constraints, i.e., time, budget, team skills, etc. $W_j$ represents the weight assigned to different actions based on the priority of that action for a particular *MF* and $n$ represents the number of actions corresponding to each $MF_i$.

---

**Algorithm 1:** Working of the proposed Methodology

---

Let MF denote mediating factors; $A_1, A_2, A_3 \ldots \ldots A_n$ are actions to promote the MF; SPV refers to SAP verification,

---

**Begin**

    *Choose SSDLC*

    *Establish Requirements (SSDLC)*

    *Identify MFi for SPV*

                *for (i = 1 to 10)*

        {

Identify $A_1, A_2, A_3 \ldots \ldots A_n \rightarrow MF_i$

                *Prioritize* $(A_1, A_2, A_3 \ldots \ldots A_n)$

        *Assignweights* $(A_1, A_2, A_3 \ldots \ldots A_n)$

        *Evaluate* $(A_1, A_2, A_3 \ldots \ldots A_n) \rightarrow MF_i$

        }

      *If* $(A_1, A_2, A_3 \ldots \ldots A_n) = true$

         *Go to End*

      *else*

        {

          *Identify missing* $A_i$

           *evaluateimpact* $(A_i) \rightarrow MF_i$

        }

      *If impact = high*

         *address* $A_i$

      *else*

         *Ignore* $A_i$

**End**

---

The accumulative *SPV* can be computed as mentioned in Equation (6):

$$Acc(SPV) = (\frac{1}{n} \sum_{j=1}^{5} A_j W_j + \frac{1}{n} \sum_{j=1}^{5} A_j W_j + \frac{1}{n} \sum_{j=1}^{6} A_j W_j + \frac{1}{n} \sum_{j=1}^{4} A_j W_j + \frac{1}{n} \sum_{j=1}^{3} A_j W_j + \frac{1}{n} \sum_{j=1}^{6} A_j W_j + \frac{1}{n} \sum_{j=1}^{7} A_j W_j$$
$$+ \frac{1}{n} \sum_{j=1}^{3} A_j W_j + \frac{1}{n} \sum_{j=1}^{4} A_j W_j \frac{1}{n} \sum_{j=1}^{6} A_j W_j)) \geq Acc(Tv) \tag{6}$$

To evaluate the impact of a particular $MF_i$ on *SPV*, the partial derivatives can be found w.r.t. a particular $MF_i$. Equation (7) represents the impact of a specific $MF_i$ on *SPV* by taking the partial derivatives of Equation (5), as given in Equation (7):

$$\frac{\partial}{\partial MF_i}(SPV) = \frac{\partial}{\partial MF_i}(MF_1, MF_2, MF_3 \ldots \ldots MF_{10}) \text{ where } 1 \leq i \leq 10 \tag{7}$$

For optimization, a threshold value is needed to determine whether the optimization is achieved or not. The comfort index, $C_i$, in this case is calculated using the formula mentioned in Equation (8):

$$C_i = \sum_{i=1}^{10} \left( Tv_i - \left( \frac{e_i}{MF_i} \right)^2 \right) \tag{8}$$

where $e_i$ represents the differences between the actual values and estimated values of $MF_i$.

To evaluate the impact of a particular MF, software organizations need to assign it a weight first. The weight assignment depends on the MF's importance for a particular project, the time used to implement that MF, and the expected ROI. For example, to accomplish $MF_1$, an organization needs to take five actions. Next, all these actions need to be prioritized, and each action should be assigned a weight based on its priority.

## 6. Evaluation of Proposed Methodology Using Case Study

Once the MFs for SAP verification are defined along with actions to promote it, there is a need to evaluate the suggested MFs and promoting actions. To do so, we performed two case studies in real settings. A case study is an effective assessment method that gives sufficient information about a real-world situation [39]. Because the proposed MFs are for the software sector, a case study was an acceptable research approach in this study. We performed two case studies in this study to assess the suggested technique. The primary goal of these case studies was to demonstrate that the discovered MFs and related actions may be employed in a real-world setting and to demonstrate the feasibility of employing these actions to enhance SAP verification.

To carry out the case study, we talked with personnel from various software companies, informed them about the identified practices, and encouraged them to participate in our research. They were instructed to undertake these activities to examine their SAP verification methods using the six-scale evaluation mentioned in Table 4. These evaluation scales are already used in existing reputable studies [40,41] for the evaluation of their proposed approach. The respondents involved in both case studies completed the assessment at their office and emailed the findings and comments to us. We evaluated the received information using the formula mentioned in Equation (5). The same evaluation method has also been adopted by researchers in existing scientific studies [6,42,43].

**Table 4.** Definition of evaluation scales.

| Scales | Definition |
| --- | --- |
| 0 (poor) | Management does not feel the need to implement/adapt MFs and their corresponding practices for SAP verification |
| 2 (weak) | Management realized the importance of adapting MFs and their corresponding practices for SAP verification |
| 4 (Fair) | Management has defined a plan for the adaptation of MFs and their corresponding practices for SAP verification |
| 6 (marginally qualified) | MFs and their corresponding practices for SAP verification are used for some projects |
| 8 (Qualified) | Management has integrated the MFs and their corresponding practices in the SAP verification process |
| 10 (Expert) | Management is fully committed to using MFs and their corresponding practices for SAP verification throughout the organization |

### 6.1. Results of an Analysis for an Organization A

Organization A is an IT firm established in Pakistan that operates in 14 countries throughout Asia, Oceania, Europe, and North America. It is one of the few top IT firms offering comprehensive services to organizations by combining cutting-edge methods with in-depth technological research and the expertise of industry leaders. Its customers vary from startups to Fortune 500 companies and government agencies, and it serves them all with cutting-edge technology solutions in the realm of new media. Its services are reliable, safe, and of high quality, allowing businesses of all kinds to flourish. With the goal of growing its company and establishing itself as one of the top web development firms, it partners with other established businesses.

Organization A offers a wide range of services, including website design and development, mobile and desktop app development, enterprise software development, digital marketing and e-commerce solutions, information technology and research-based consultancy, SharePoint development, user interface design, optimization, and search engine optimization. The respondents of the case study were asked to first rank/weight each MF's action based on the importance of that action for the organization; the value of weight was from 0–1, where one means highly important and zero refers to "not at all important".

Once the weight was assigned to each action corresponding to each MF, the respondents were further asked to evaluate these actions according to its use in their organization (as mentioned in Table 4). If the management of the organization was fully committed to using MFs and their corresponding practices for SAP verification throughout the organization, then the evaluation score was 10, otherwise it varied depending on its implementation in the organization. Once all the actions were assigned weights and evaluated by the participants, the weighted average of each MF was calculated using the formulae mentioned in Equation (5). Table 5 shows the results of MF1 as evaluated by the respondents of Organization A for a better understanding of the proposed methodology. "W" in Table 5 refers to the average value of the weights assigned to each action by all the respondents, and "X" refers to the average evaluation by all the respondents for a particular action.

**Table 5.** MF1 evaluation by Organization A.

| MF1 | A1 | A2 | A3 | A4 | A5 |
|-----|-----|-----|-----|-----|-----|
| W | 1 | 0.9 | 1 | 0.9 | 0.8 |
| X | 9 | 10 | 8 | 8 | 8 |
| WX | 9 | 9 | 8 | 7.2 | 6.4 |

Evaluation (MF1) = $\frac{\sum_{i=1}^{5} w_i x_i}{5} = 39.6/5 = 7.92 \cong 8$

Figure 4 shows the assessment results of Organization A for each MF and its corresponding practices.

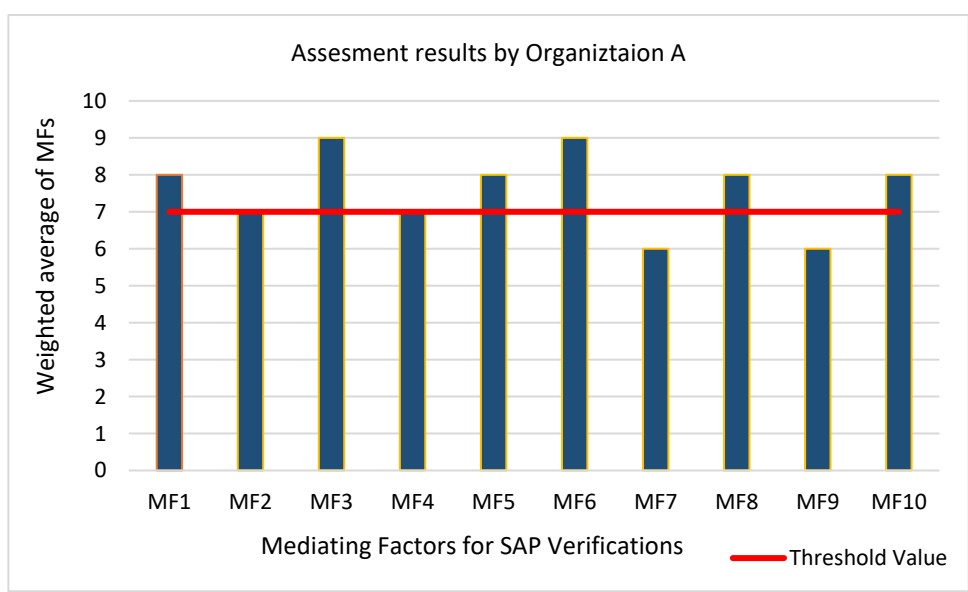

**Figure 4.** Assessment results by Organization A for the identified MFs.

The findings/key points identified during the assessment process were as follows:

- The company is committed to implementing SAP's MFs. It has already implemented most of the specified practices since the computed values for most of the MFs were greater than or equal to the threshold value.
- Respondents said that the customer is usually unaware of the security needs of the product, due to which documenting the SAP verification requirement is a challenge.
- The organization is deficient in MF7 and MF9, indicating that management should offer tools for SAP verification and provide time and resources for SAP verification.

### 6.2. Results of an Analysis for an Organization B

Organization B is a frontrunner in the software engineering industry. It provides startups, small- and medium-sized businesses, and enterprises with top-tier custom software development services. It has more than 200 employees and is considered among the well-known companies in Pakistan. This company has exercised caution and forethought in disclosing the material because of the importance of the industry to the economy and the special nature of the information at hand.

Organization B presented us with its assessment of MFs and related actions, as illustrated in Figure 5. The responders from Organization B were given identical instructions to those given to Organization A. The weighted total of actions for each MF was calculated using the formulae in Equation (5). Each MF had the same threshold value specified. The important findings derived from organization B's responses are described below.

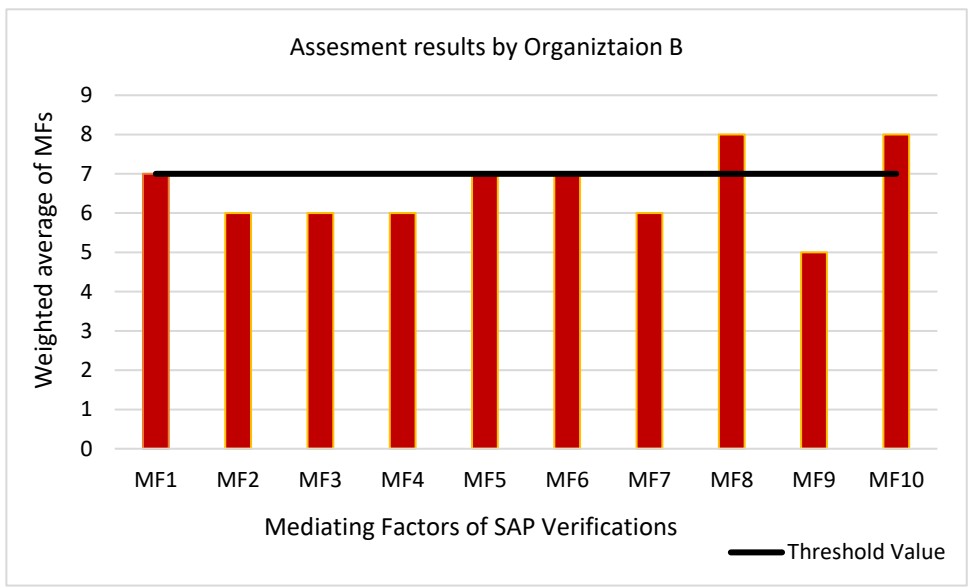

**Figure 5.** Assessment results by Organization B for the identified MFs.

- Most of the MFs have not been adapted by Organization B, as the weighted sum of most of these MFs was less than the threshold values.
- The respondents told us that organizations do not provide tool support, time and budget allocation, and training to the employees while they expect secure and quality products.
- SAP requirements are not documented properly, which hinders the process of SAP verification.
-  The customer is not actively involved in documenting and testing security requirements.
- There is a need to adapt the mentioned practices to promote SAP culture in the organization.

### 6.3. Case Study Participants' Recommendations/Feedback

The case study allowed us to assess the applicability of the suggested SAP verification practices in a real-world setting. A summary of the responses received from case study participants is as follows

- The participants were able to understand the proposed methodology without any assistance; they only asked for an explanation of some of the mentioned practices
- The mentioned MFs and their corresponding actions helped them to assess the current SAP verification level of their organizations
- The management team showed interest in implementing the mentioned practices in their organizations and considered them suitable for achieving a secure and quality product.

### 6.4. Findings and Implications

The results from both case studies showed that the identified MFs and their corresponding actions were applicable in the software industry, and medium-sized organizations (such as Organization A in our case) are already using these actions. If we analyze the statistics of Figure 4, eight out of ten practices have already been used by Organization A at an optimum, as the aggregated value of the MF was more than the set threshold value. Organization B also uses all the MFs and their corresponding actions, although not at an adequate level. There is a need to provide awareness about these MFs and complementary measures to organizations so that they can incorporate security and performance in their developed products. This study will provide them with a roadmap to address SAP verification issues.

### 6.5. Threats to Validity

This section discusses the challenges to the validity of our investigation and how they might be mitigated to boost confidence in the findings of our study.

#### 6.5.1. Construct Validity

An essential limitation of the proposed methodology was assigning the practices/actions to the MF. We completed the task based on our own experience and the knowledge gathered from the literature research. To ensure construct validity, the MFs and their corresponding practices were evaluated iteratively by two university professors with sufficient expertise in teaching a software SAP course.

#### 6.5.2. External Validity

It cannot be assured that the findings and conclusions of this research will be relevant to all software development businesses. Specifically, we performed case studies in just two firms to evaluate our suggested technique. As a result, generalizations about the applicability of the suggested approach should be made with care when drawing results.

#### 6.5.3. Internal Validity

A fair picture of the current procedures utilized to enhance SAP verification was gathered via a literature review, which yielded MFs and the actions that corresponded with them. Because some sources lacked adequate or clear information on our MFs, objectivity was required in selecting sources and extracting data. The sources were thoroughly vetted and chosen based on a set of quality criteria to counteract this danger. We are worried about the veracity of the findings since the participants in our case study assessed the organization's procedures. The evaluation may be subjective since the assessor must pay close attention to the given criteria to obtain accurate findings. Participants' prior knowledge of software SAP and their position within the company are also important considerations.

## 7. Conclusions and Future Work

The main goal of this research was to identify the MFs that improve software SAP. Once the MFs were identified, the next step was to identify the actions that might help to achieve these MFs. To identify MFs and their corresponding actions, both the formal and grey literature were used. The identified MFs and their promoting actions were modeled mathematically, and a case study with two small- and medium-sized organizations was conducted to validate the proposed methodology. The outcomes of these studies confirmed the usability of our proposed approach in a real-world environment. This work is expected to help software organizations increase the SAP verifications of their products and improve their SAP verification process. They can use the suggested MFs to assess the maturity of their SAP verification process. The study findings will also increase developers' level of awareness about SAP verification. The findings of this research will be relevant for

all software development businesses as they have the liberty to weight/rank the actions corresponding to each MFs according to the importance for their organization.

This study may be improved in several ways. Researchers may pursue the following open research directions:

- This approach might be tailored to the demands of various organizations based on their facilities and methods.
- The proposed technique might be modified to include special features linked to the technologies such as IoT, big data, Blockchain, and cloud computing.
- The suggested technique might be published publicly and updated with new academic and industrial practices.
- The evaluation process might be automated in the future to reduce the burden from user's shoulders and to remove biasedness.
- This study considered SAP as a set of unrelated metrics, which is not always correct. SAP, by its nature, tends to bear contradictions. For instance, the more cryptographically secure a software is, the less its performance metrics' values. In the future, this issue may be addressed.
- Further evaluation cases may be added to strengthen the findings.

**Author Contributions:** Conceptualization, M.F.A. and M.H.; Data curation, M.F.A. and M.H.; Formal analysis, M.F.A. and M.H.; Methodology, M.F.A. and M.H.; Supervision, M.H.; Writing—original draft, M.F.A. and M.H.; Writing—review and editing, M.F.A. and M.H. All authors have read and agreed to the published version of the manuscript.

**Funding:** This work was funded by the Deanship of Scientific Research at Jouf University under grant No (DSR2022-NF-03).

**Institutional Review Board Statement:** Not applicable.

**Informed Consent Statement:** Not applicable.

**Data Availability Statement:** Will be furnished on request.

**Conflicts of Interest:** The authors declare no conflict of interest.

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
