# Peer review of "Improving the Safety and Security of Software Systems by Mediating SAP Verification"

_applsci, doi:10.3390/app13010647_

Round 1
Reviewer 1 Report
This paper has a good motivation that aims to optimize SAP verification by identifying and analyzing the factors (i.e., MFs). Meanwhile, this paper proposes a prioritization framework to quantify the impact of each MF and its corresponding actions. However, I still have some minor concerns about this paper.
1) How the authors summarize the corresponding actions? Are they also from the literature review or proposed by the authors? And how the authors guarantee the correctness and effectiveness of these actions? The authors may clarify them.
2) There are two many duplicate formulas in this paper. Such as Formula (5)-(14) and (16)-(25). Even though they are designed for different MFs, they are repeated semantically. The authors may delete them. Also, the authors may need to unify the format of all formulas, especially their their alignment。
3)Line 375: Table 1 --> Table 2
Author Response
General View: This paper has a good motivation that aims to optimize SAP verification by identifying and analyzing the factors (i.e., MFs). Meanwhile, this paper proposes a prioritization framework to quantify the impact of each MF and its corresponding actions. However, I still have some minor concerns about this paper.
Author response: Thank you very much for your encouraging remarks, valuable feedback, and efforts. Here is the point-wise response to all comments.
Reviewer Concern # 1: How do the authors summarize the corresponding actions? Are they also from the literature review or proposed by the authors? And how the authors guarantee the correctness and effectiveness of these actions? The authors may clarify them.
Author Response: The mediating factors(MFs) and corresponding actions are taken from existing literature (we also added this detail in the updated manuscript and highlighted it). The following papers are considered for selecting mediating factors and corresponding actions. Some of the selected MFs and corresponding actions were common in all these studies, while a few of them were only repeated twice or thrice. We combined all these MFs and corresponding actions so that we could provide a comprehensive view to the software practitioners. The guarantee about the correctness and effectiveness of these actions is that they are extracted from reputable published work. The below-mentioned papers are also there in the manuscript references list.
- Ribeiro, V.V., Cruzes, D.S. and Travassos, G.H., 2018, November. A perception of the practice of software security and performance verification. In 2018 25th Australasian Software Engineering Conference (ASWEC)(pp. 71-80). IEEE.
- Ribeiro, V.V., Cruzes, D.S. and Travassos, G.H., 2022. Moderator factors of software security and performance verification. Journal of Systems and Software, 184, p.111137.
- Ribeiro, V., Cruzes, D.S. and Travassos, G.H., 2020, December. Understanding Factors and Practices of Software Security and Performance Verification. In Anais Estendidos do XIX Simpósio Brasileiro de Qualidade de Software(pp. 11-20). SBC.
- Vidigal Ribeiro, Victor, Daniela Soares Cruzes, and Guilherme Horta Travassos. "Moderator Factors of Software Security and Performance Verification." arXiv e-prints(2021): arXiv-2102.
Reviewer Concern # 2: There are two many duplicate formulas in this paper. Such as Formula (5)-(14) and (16)-(25). Even though they are designed for different MFs, they are repeated semantically. The authors may delete them. Also, the authors may need to unify the format of all formulas, especially their their alignment
Author Response: We have deleted the repeated formulas and merged them into a single general formula. We also unified the format and alignment of all the equations in the updated manuscript.
Reviewer Concern # 3: Line 375: Table 1 --> Table 2
Author Response: Table 1 is the list of abbreviations used in the study (see Line 88) while table 2 is MFs and the corresponding actions to promote these MFs (see Line 376)
Reviewer 2 Report
The abstract can be rewritten to clearly portray the research goals. "Verification efforts should be integrated into the development process so that related problems may be discovered and deployment failures can be avoided.
Related problems- What are these
Deployment failures - There are many types of deployment failures, and the authors need to be precise on what they are trying to address.
Author Response
Reviewer Concern # 1: The abstract can be rewritten to clearly portray the research goals. "Verification efforts should be integrated into the development process so that related problems may be discovered and deployment failures can be avoided.
Author response: Thank you very much for your valuable time and efforts. The abstract is completely rewritten in the updated manuscript to make it more precise and clear, below is the updated abstract
Security and performance (SAP) are two critical NFRs that affect the successful completion of software projects. Organizations need to follow the practices that are vital to SAP verification. These practices must be incorporated into the software development process to identify SAP related defects and avoid failures after deployment. This can be achieved only if organizations are fully aware of SAP verification activities and appropriately include them in the software development process. However, there is a lack of awareness of the factors that influence SAP verification, which makes it difficult for businesses to improve their verification efforts and ensure that the released software meets these requirements. To fill this gap, this research study aims to identify mediating factors (MFs) influencing SAP verification and actions to promote them. Ten MFs and corresponding actions were identified after thoroughly reviewing the existing literature. The mapping of MFs and corresponding actions were initially evaluated with the help of a pilot study. Mathematical modeling was utilized to model these MFs and examine each MF's unique effect on software SAP verification. In addition, two case studies with a small and a medium-sized organization were used to better understand the function these MFs play in the process of SAP verification. The research findings suggest that MFs assist software development organizations in their efforts to integrate SAP verification procedures into their standard software systems. Further investigation is required to support the understanding of these MFs when building modern software systems.
Reviewer Concern # 2: Related problems- What are these
Author response: "Related problems" refer to the issues which occur in the absence of security and performance verification process
Reviewer Concern # 3: Deployment failures - There are many types of deployment failures, and the authors need to be precise on what they are trying to address.
Author response: By deployment failures, we mean that if the security and performance verification activities are not performed during development, problems may occur after software deployment.
The above-mentioned comments may be due to the writing style, so we have rewritten the complete abstract in a precise way.
A complete proof-read of the paper is also done to avoid typos and grammatical issues.
Reviewer 3 Report
The paper's primary goal is to develop the methodology of security and performance (SAP) mediating factors (MFs) for secure software development. The paper's main strengths are the topical subject, the materials' clear structure, and the clarity of the content.
The main weakness of the paper is related to the lack of proof for the received results' adequacy and the adequacy of their interpretation. In addition, the methodology is very generic. It moves almost all the tasks and all responsibility for the result on the users' shoulders, while it may bring even more 'human factor' into yielded results when it wasn't used. The standard practice of such methods is to determine what to measure (i.e., clear metrics, for instance, one may refer to NIST 800-53) and how to do this (for example, one may refer to NIST's Cybersecurity Framework). Thus there is no proof that in the case when the methodology's user follows the methodology, he/she will succeed in reaching the goal.
General concept comments:
1. Proposed MFs (table 1) contain generic actions that were proir mentioned. However, those mentionings didn't cover proof of efficiency, completeness for both MFs and actions, or even the appropriateness of these actions' application order. The reader may consider this as unfounded/groundless. Meanwhile, the authors state that certain analyses were performed. Therefore, it is unclear why such crucial for scientific perception results weren't delivered.
2. The methodology lacks reasoning and /or proof of the proposed solution's appropriateness. For example, no arguments were provided for the proposed scales used for the actions performance estimation (table 3). Furthermore, there were no descriptions and proof of adequacy for the weight values applied in the case study example. It should be clear whether weight values are taken from the range of [0; 1] and how one should determine them. Therefore a lot of unanswered questions arose, such as:
2.1 Will the methodology work if all weights are less than 1 (all are less than 0,1, for example)?
2.2 How do we interpret the result, which may never reach 7 because of such weight distribution (consider A1, A2, ... all 10 for the case)?
2.3 Why was 7 chosen as a threshold?
2.4 Why does the threshold not depend on the weights?
2.5 Is it correct to sum the received estimations?
...
Thus, despite great potential, the paper seems raw and lacks scientific consistency.
3. Authors consider SAP as a set of unrelated metrics, which is not always correct. SAP, by its nature, tends to bear contradictions. For instance, the more cryptographically secure software is the less its performance metrics' values. To illustrate the point, in particular, one may refer to table 4 for MF1. Consider case A1= A2 = 10, A3 = 2, A4= A5=10 (everything that is defined by the methodology is one perfectly, i.e. nothing due to the absence of the latter at the enterprise), which means a lack of verification methodology (management is aware that this is important but does nothing to address the issue of absent verification methodology - according to table 3). Nevertheless, this case will yield a value close to the value 8, which means MF1 is fine, and there is no reason to worry - the same as in case 1, while in the case, there is no systematic methodology at all.
Specific comments:
4. Lines 81-84 mix Arabic and Latin numbers while enumerating sections of the paper.
5. Figures 1 and 2 contain elements focused on emotions (e.g., `quality`, safe hands, etc.), while for a scientific paper, it is better to address logic and stay emotionally neutral.
6. Incomplete sentences were met in lines 216 and 321.
7. Formulas (5)-(14) are similar and lead the reader away from the main point - it would be better to either use one generic formula or state that MF2-MF10 are determined similarly.
8. Formula (3) is not clear. The functional relation between SPV and MF1, ... , MF10 was not proved. Therefore, formulas (16)-(25) seem questionable.
9. In formula (26) authors use error value, while they didn't mention the way how the actual value of parameters can be yielded in practice to compute the error.
10. It is stated that "...a case study with two small and medium-sized organizations was conducted..." (line 610), while in the abstract, it is stated, "...two case studies with a big and a medium-sized organization were used...", which contradicts the former one.
Author Response
General view: The paper's primary goal is to develop the methodology of Security and performance (SAP) mediating factors (MFs) for secure software development. The paper's main strengths are the topical subject, the materials' clear structure, and the clarity of the content.
Author response: Thank you very much for your encouraging remarks, valuable feedback, and efforts. Here is the point-wise response to all comments.
Reviewer Concern # 1: The main weakness of the paper is related to the lack of proof for the received results' adequacy and the adequacy of their interpretation. In addition, the methodology is very generic. It moves almost all the tasks and all responsibility for the result on the users' shoulders, while it may bring even more 'human factor' into yielded results when it wasn't used. The standard practice of such methods is to determine what to measure (i.e., clear metrics, for instance, one may refer to NIST 800-53) and how to do this (for example, one may refer to NIST's Cybersecurity Framework). Thus there is no proof that in the case when the methodology's user follows the methodology, he/she will succeed in reaching the goal.
Author Response: We appreciate your suggestion and consider the methodology generic. However, as software is developed by humans and for humans, therefore, most of the interpretation is done by the development team. We plan to automate the evaluation process in the future to reduce the burden from user's shoulders and remove bias. We have mentioned this limitation in the updated manuscript and also added this in future work part (see lines 604-605) of the updated manuscript.
Further, the same evaluation criteria were also used by senior researchers (in reputable venues) in the existing studies, as given below
- Niazi, M., Saeed, A.M., Alshayeb, M., Mahmood, S. and Zafar, S., 2020. A maturity model for secure requirements engineering. Computers & Security, 95, p.101852.
- Niazi, Mahmood, Mohamed El-Attar, Muhammad Usman, and Naveed Ikram. "An empirical study identifying high perceived value requirements engineering practices in global software development projects." In Proceedings of the 7th International Conference on Software Engineering Advances (ICSEA), Lisbon, pp. 283-288. 2012.
- Humayun, M., Jhanjhi, N.Z., Almufareh, M.F. and Khalil, M.I., 2022. Security Threat and Vulnerability Assessment and Measurement in Secure So
We have added the above references to support our evaluation methodology in the updated manuscript.
Reviewer Concern # 2: Proposed MFs (table 1) contain generic actions that were proir mentioned. However, those mentionings didn't cover proof of efficiency, completeness for both MFs and actions, or even the appropriateness of these actions' application order. The reader may consider this as unfounded/groundless. Meanwhile, the authors state that certain analyses were performed. Therefore, it is unclear why such crucial for scientific perception results weren't delivered.
Author Response: The mediating factors (MFs) and corresponding actions are taken from existing literature (we also added this detail in the updated manuscript and highlighted it). The following papers are considered for selecting mediating factors and corresponding actions. Some of the selected MFs and corresponding actions were common in all these studies, while a few of them were only repeated twice or thrice. We combined all these MFs and corresponding actions so that we could provide a comprehensive view to the software practitioners. The guarantee about the correctness and effectiveness of these actions is that they are extracted from reputable published work. The below-mentioned papers are also there in the manuscript references list.
- Ribeiro, V.V., Cruzes, D.S. and Travassos, G.H., 2018, November. A perception of the practice of software security and performance verification. In 2018 25th Australasian Software Engineering Conference (ASWEC)(pp. 71-80). IEEE.
- Ribeiro, V.V., Cruzes, D.S. and Travassos, G.H., 2022. Moderator factors of software security and performance verification. Journal of Systems and Software, 184, p.111137.
- Ribeiro, V., Cruzes, D.S. and Travassos, G.H., 2020, December. Understanding Factors and Practices of Software Security and Performance Verification. In Anais Estendidos do XIX Simpósio Brasileiro de Qualidade de Software(pp. 11-20). SBC.
- Vidigal Ribeiro, Victor, Daniela Soares Cruzes, and Guilherme Horta Travassos. "Moderator Factors of Software Security and Performance Verification." arXiv e-prints(2021): arXiv-2102.
Reviewer Concern # 3: ) The methodology lacks reasoning and /or proof of the proposed solution's appropriateness. For example, no arguments were provided for the proposed scales used for the actions performance estimation (table 3). Furthermore, there were no descriptions and proof of adequacy for the weight values applied in the case study example. It should be clear whether weight values are taken from the range of [0; 1] and how one should determine them. Therefore a lot of unanswered questions arose, such as:
2.1 Will the methodology work if all weights are less than 1 (all are less than 0,1, for example)?
2.2 How do we interpret the result, which may never reach 7 because of such weight distribution (consider A1, A2, ... all 10 for the case)?
2.3 Why was 7 chosen as a threshold?
2.4 Why does the threshold not depend on the weights?
2.5 Is it correct to sum the received estimations?
Thus, despite great potential, the paper seems raw and lacks scientific consistency.
Author response: Thank you for your valuable suggestion and feedback. However, the idea of weight assignment, threshold value selection, and calculating estimations and results is taken from existing published studies. Below are the details
The proposed scales used for the performance of the action estimation are taken from the following studies which are already published in well-known journals (we have also mentioned this in the updated manuscript and highlighted). The weight values are taken between [0 to 1], where 0 means unimportant and one means highly valuable.
- Al-Matouq, Hassan, Sajjad Mahmood, Mohammad Alshayeb, and Mahmood Niazi. "A maturity model for secure software design: A multivocal study." IEEE Access 8 (2020): 215758-215776.
- Niazi, Mahmood, Karl Cox, and June Verner. "A measurement framework for assessing the maturity of requirements engineering process." Software Quality Journal 16, no. 2 (2008): 213-235
10 is the highest value, so we have taken seven as a threshold, as it is above average. Further, the same threshold value has been used in the following published study
- Al-Matouq, Hassan, Sajjad Mahmood, Mohammad Alshayeb, and Mahmood Niazi. "A maturity model for secure software design: A multivocal study." IEEE Access 8 (2020): 215758-215776.
Regarding how to interpret results; we have described the evaluation of MF1 for a better understandability of overall interpretation as below (also highlighted in the updated manuscript)
Once all the actions were assigned weights and evaluated by the participants, the weighted average of each MFs was calculated using the formulas mentioned in equation 5. Table 4 shows the results of MF1 as evaluated by the respondents of organization A for a better understanding of the proposed methodology. "W" in table 4 refers to the average value of the weights assigned to each action by all the respondents, and "X" refers to the average evaluation by all the respondents for a particular action.
Table 4: MF1 evaluation by organization A
MF1 |
A1 |
A2 |
A3 |
A4 |
A5 |
W |
1 |
0.9 |
1 |
0.9 |
0.8 |
X |
9 |
10 |
8 |
8 |
8 |
WX |
9 |
9 |
8 |
7.2 |
6.4 |
Evaluation (MF1) =
Reviewer Concern # 4: ) Authors consider SAP as a set of unrelated metrics, which is not always correct. SAP, by its nature, tends to bear contradictions. For instance, the more cryptographically secure software is the less its performance metrics' values. To illustrate the point, in particular, one may refer to table 4 for MF1. Consider case A1= A2 = 10, A3 = 2, A4= A5=10 (everything that is defined by the methodology is one perfectly, i.e. nothing due to the absence of the latter at the enterprise), which means a lack of verification methodology (management is aware that this is important but does nothing to address the issue of absent verification methodology - according to table 3). Nevertheless, this case will yield a value close to the value 8, which means MF1 is fine, and there is no reason to worry - the same as in case 1, while in the case, there is no systematic methodology at all.
Author Response: We agree with your comments that Security and performance are sometimes contradictory, especially for a safety-critical system. However, this was an initial study; the aim was to provide a complete state-of-the-art about SAP and collect SAP verification practices and actions. Further, the idea of the proposed methodology evaluation was taken from existing studies in the field of software engineering [40-43]. At this stage, change in methodology is difficult as we need to conduct both case studies again. However, we added this to the updated manuscript's limitation and will try to address it in the future (see lines 608-611 in the revised manuscript).
Reviewer Concern # 5: ) Lines 81-84 mix Arabic and Latin numbers while enumerating sections of the paper.
Author Response: We have updated the numbers and used a consistent style
Reviewer Concern # 6: ) Figures 1 and 2 contain elements focused on emotions (e.g., `quality`, safe hands, etc.), while for a scientific paper, it is better to address logic and stay emotionally neutral.
Author Response: Thank you for the valuable suggestion. We updated the emotional icons in Figures 1 and 2
Reviewer Concern # 7: ) Incomplete sentences were met in lines 216 and 321.
Author Response: Incomplete sentences are corrected
Reviewer Concern # 8: ) Formulas (5)-(14) are similar and lead the reader away from the main point - it would be better to either use one generic formula or state that MF2-MF10 are determined similarly
Author Response: We have used one generic formula for equations 5-14 and 16-25 in the updated manuscript as below
(5)
Reviewer Concern # 9: ) Formula (3) is not clear. The functional relation between SPV and MF1, ... , MF10 was not proved. Therefore, formulas (16)-(25) seem questionable.
Author Response: Aaccording to formula three and the existing literature review, security and performance verification depends on the extracted MFs. However, we just modelled this relationship mathematically in equation 3 and tried to prove it with the help of a case study. Further, formulas 16-25 are merged into a single generic formula.
Reviewer Concern # 10: ) In formula (26), authors use error value, while they didn't mention the way how the actual value of parameters can be yielded in practice to compute the error.
Author Response: The actual value of the parameters will be calculated as mentioned in the process from lines (488-496). At the same time, the estimated values will be decided by the organization, which depends on the nature of the project and other constraints.
Reviewer Concern # 11: It is stated that "...a case study with two small and medium-sized organizations was conducted..." (line 610), while in the abstract, it is stated, "...two case studies with a big and a medium-sized organization were used...", which contradicts the former one.
Author Response: There was a typo; we have updated it accordingly
Reviewer 4 Report
The purpose of this research is to identify the Mediating Factors (MFs) that have an impact on software systems' SAP. These MFs were modelled mathematically, and each MF's particular impact on software SAP verification was examined. Adequate background for Software security verification and Software performance verification. The related works are analysed critically and the paper contribution has defined clearly. The proposed methodology was evaluated by using two case studies of two IT Companies. However, it is unclear how the proposed approach has been implemented. In the Evaluation Section 5, it is unclear how the assessments are conducted and how the findings are generated.
I would recommend separating off section 3.1, "Mediating Factors for SAP Verification," and placing it in its own section rather than being included in the related works. It might be better to represent the equations differently on pages 12 and 13. On page 9 and 12, I think Table 2 should be Table 1 instead since there is no Table 2 in the paper. Table 3 should come after Table 2 in the proper sequence, and the remaining tables should follow suit. The paper needs to be edited.
Author Response
General view: The purpose of this research is to identify the Mediating Factors (MFs) that have an impact on software systems' SAP. These MFs were modelled mathematically, and each MF's particular impact on software SAP verification was examined. Adequate background for Software security verification and Software performance verification. The related works are analysed critically and the paper contribution has defined clearly. The proposed methodology was evaluated by using two case studies of two IT Companies.
Author response: Thank you very much for your encouraging remarks, valuable feedback, and efforts. Here is the point-wise response to all comments.
Reviewer Concern # 1 ) However, it is unclear how the proposed approach has been implemented. In the Evaluation Section 5, it is unclear how the assessments are conducted and how the findings are generated.
Author Response: As far as the implementation of proposed approach is concerned, we evaluated the approach based on the existing process used by the organizations involved in the case study to know the importance of extracted SAP MFs and corresponding actions. The same evaluation strategy has also been used in existing published research studies mentioned below
- Al-Matouq, Hassan, Sajjad Mahmood, Mohammad Alshayeb, and Mahmood Niazi. "A maturity model for secure software design: A multivocal study." IEEE Access 8 (2020): 215758-215776.
- Niazi, Mahmood, Karl Cox, and June Verner. "A measurement framework for assessing the maturity of requirements engineering process." Software Quality Journal 16, no. 2 (2008): 213-235
The process of conducting assessment and generating finding is further elaborated in the revised manuscript and is highlighted (see lines 488-496)
Reviewer Concern # 2 ) I would recommend separating off section 3.1, "Mediating Factors for SAP Verification," and placing it in its own section rather than being included in the related works. It might be better to represent the equations differently on pages 12 and 13. On page 9 and 12, I think Table 2 should be Table 1 instead since there is no Table 2 in the paper. Table 3 should come after Table 2 in the proper sequence, and the remaining tables should follow suit. The paper needs to be edited.
Author Response: we have separated off section 3.1 and numbered it section 4 in the revised manuscript.
We have also updated equations and merged similar equations into a generic form. We have used generic formulas for equations 5-14 and 16-25 in the updated manuscript
Table 1 is the list of abbreviations used in the study (see Line 88), while table 2 is MFs and the corresponding actions to promote these MFs (see Line 376)
A complete proof-read of the paper is also done to check for typos and grammatical issues
Reviewer 5 Report
Background. In my opinion, this section is not well focused, since an SSDLC includes the security verification techniques that are mentioned and others that are not mentioned, such as functional security tests for validation of security requirements or the use of interactive tools (IAST). In addition, it is necessary to address the threat and risk modeling techniques that allow establishing security requirements. The astract and introduction must be also adapted.
Related work. The investigation of the state of the art should be more complete in terms of analyzing existing SSDLC models and comparing them in order to choose the most appropriate one.
Methodology. It should contemplate the choice of an SSDLC model establishing the requirements for it. In itself, an SSDLC model defines the process of software security verification. In my opinion, the proposed methodology can be complementary to the chosen SSDLC to identify defects in its implementation. Based on this, the concept of the proposed methodology should be adapted.
Evaluation cases must be adapted
Author Response
Reviewer Concern # 1: Background. In my opinion, this section is not well focused, since an SSDLC includes the security verification techniques that are mentioned and others that are not mentioned, such as functional security tests for validation of security requirements or the use of interactive tools (IAST). In addition, it is necessary to address the threat and risk modeling techniques that allow establishing security requirements. The abstract and introduction must be also adapted.
Author response: Thank you very much for your valuable time and efforts. Below is the line-by-line response to each comment
- The abstract is completely rewritten to make it more precise and clear.
- The introduction section is also revised in light of comments.
- The use of IAST has been added in the updated manuscript (see Lines 161-165). We also updated Figure 1 by adding the technique of IAST.
- We have added a threat modeling process in section 2.1.2 using text (See lines 132-136) and Figures (See Figure 2)
Reviewer Concern # 2: Related work. The investigation of the state of the art should be more complete in terms of analyzing existing SSDLC models and comparing them in order to choose the most appropriate one.
Author response: Thank you very much for your valuable time and efforts. We have completed the investigation of state of the art by analyzing and comparing existing SSDLC models (See Lines 285-323). The following text has been added in the updated manuscript
3.1. Existing SSDLC models
In order to better understand the SSDLC, we need to analyze the existing SSDLC models. Below we provide the details of common SSDLC models and their comparison.
3.1.1. SSE-CMM
Based on the Software Engineering CMM, SSE-CMM is a process-oriented paradigm used to create safe systems. The SSE-CMM is divided into processes and stages of maturity. In general, the procedures specify what the security engineering process must do, and the maturity levels classify how well the process achieves its objectives. There are 11 process areas in it, to use the SSE-CMM, current processes are simply analyzed to see whether basic processes have been satisfied and what maturity levels they have attained. The same procedure may assist a company in identifying security engineering procedures that they may need but do not already have in place.
3.1.2. Microsoft SDL
Microsoft's SDL incorporates stringent security standards, industry-specific tools, and required procedures into creating and maintaining all software products. By following the SDL methods and criteria, the development teams may utilize MS-SDL to produce better secure software with fewer and less serious vulnerabilities at a lower development cost. There are seven phases in MS-SDL, comprising the five SDLC core stages and two auxiliary security tasks. To make sure all security and privacy needs are correctly met, each of these steps includes necessary checks and approvals. To make sure the core phases are implemented correctly and software is kept safe after deployment, the two supporting security activities, training, and response, are carried out before and after the core phases, respectively.
3.1.3. SAMM
SAMM is an open framework designed to assist businesses in developing and implementing a software security strategy appropriate to the unique threats they face. SAMM's tools may be used to assess an organization's current software security procedures, create a well-rounded software security program in measurable steps, verify the efficacy of a security assurance program, and quantify security-related endeavors. Because of its adaptability, SAMM may be used by companies of all sizes and in any development methodology. This model's flexibility means it may be used throughout an enterprise, inside a business unit, or even on a standalone project.
After discussing the three commonly used existing SSDLC models, below we provide the comparison of these models in Table 2
Security practices |
SSE-CMM |
MS-SDL |
SAMM |
Physical Security |
Yes |
No |
No |
Logical Security |
Yes |
No |
No |
Definition of Security Requirements |
Yes |
Yes |
Yes |
Secure Configuration Management |
Yes |
No |
No |
Following all applicable laws, policies, and procedures |
Yes |
No |
Yes |
Threat Modeling |
Yes |
Yes |
Yes |
Risk Analysis |
Yes |
Yes |
Yes |
Security Architecture |
Yes |
Yes |
Yes |
Security Training and Awareness |
Yes |
Yes |
Yes |
Secure Design |
Yes |
Yes |
Yes |
Source Code Analysis |
No |
Yes |
Yes |
Vulnerability Analysis |
Yes |
Yes |
Yes |
Security Verification |
Yes |
Yes |
Yes |
Vulnerability Management |
Yes |
Yes |
Yes |
Secure Development Techniques and Applications |
Yes |
Yes |
Yes |
Security in an Active Operating Environment |
Yes |
Yes |
Yes |
Secure Integration with Peripheral |
Yes |
Yes |
Yes |
Secure Delivery |
Yes |
No |
Yes |
The analysis of the literature review and comparison of existing security models show that security is one of the important concerns discussed by many researchers and security organizations. However, we didn't find studies that provide solutions for both security and performance or uncover the MFs that affect SAP.
Reviewer Concern # 3: Methodology. It should contemplate the choice of an SSDLC model establishing the requirements for it. In itself, an SSDLC model defines the process of software security verification. In my opinion, the proposed methodology can be complementary to the chosen SSDLC to identify defects in its implementation. Based on this, the concept of the proposed methodology should be adapted.
Author response: Thank you very much for your valuable time and efforts. We have incorporated the mentioned suggestions in the methodology part of the updated manuscript (See lines 426-433 and Algorithm 1)
Reviewer Concern # 4: Evaluation cases must be adapted
Author response: Thanks for the valuable suggestion. We have performed two case studies with small and medium-sized organizations to evaluate the performance of the proposed methodology using the mentioned evaluation process. Other than presenting results from both case studies, we have also provided study participants' recommendations/feedback, Findings and implications, and the possible validity threats. In the future, we plan to add more specific cases to further evaluate the proposed methodology (See lines 633-639).
Round 2
Reviewer 2 Report
There is some improvement with the changes made. I would suggest to accept the same.
Author Response
English language and style are fine/minor spell check required
Author response: Thank you very much for your time and efforts. We have done the proof-read of complete paper again for spell check
There is some improvement with the changes made. I would suggest to accept the same.
Author response: Thank you very much for your encouraging remarks, valuable feedback, and efforts.
Reviewer 5 Report
Authors have improved the paper according comments.